# Soil Organic Carbon Changes for Croplands across China from 1991 to 2012

Wentian He [1,2,3], Ping He [2,*], Rong Jiang [2,*], Jingyi Yang [4], Craig F. Drury [4], Ward N. Smith [3], Brian B. Grant [3] and Wei Zhou [2]

1    Institute of Plant Nutrition and Resources, Beijing Academy of Agriculture and Forestry Sciences, Beijing 100097, China; wentian_he@hotmail.com
2    Key Laboratory of Plant Nutrition and Fertilizer, Ministry of Agriculture and Rural Affairs/Institute of Agricultural Resources and Regional Planning, Chinese Academy of Agricultural Sciences, Beijing 100081, China; zhouwei02@caas.cn
3    Ottawa Research and Development Centre, Agriculture and Agri-Food Canada, 960 Carling Avenue, Ottawa, ON K1A 0C6, Canada; ward.smith@canada.ca (W.N.S.); brian.grant@canada.ca (B.B.G.)
4    Harrow Research and Development Centre, Agriculture and Agri-Food Canada, 2585 County Road 20, Harrow, ON N0R 1G0, Canada; jingyi.yang@canada.ca (J.Y.); craig.drury@canada.ca (C.F.D.)
*    Correspondence: heping02@caas.cn (P.H.); rong_jiang@outlook.com (R.J.)

**Abstract:** Accurate estimates of soil organic carbon (SOC) are critical for evaluating the impacts of crop and nutrient management practices on soil sustainability and global climate change. Temporal and spatial variations in topsoil (0–0.20 m) SOC were analyzed using 43,743 soil samples in China's croplands. The soil database in our study was collected from the International Plant Nutrition Institute (IPNI) China Program. The results showed an increasing trend in SOC density (SOCD) for both grain and cash crops from 1991 to 2012. The average SOCD increased by 16.8, 17.4, 11.8 and 8.7% in the north central, northwest, southeast and southwest regions, respectively, whereas average SOCD decreased by 1.3% for the northeast region between the 1991–2001 and 2002–2012 periods. For both grain and cash crops, the SOCD frequency distribution (%) increased in the ranges of 10–20, 20–30 and 30–40 Mg C ha$^{-1}$ from the 1991–2001 to the 2002–2012, but decreased in the ranges of 0–10 and 50–60 Mg C ha$^{-1}$. Additionally, SOCD increased in most major soil types across China's cropland regions, except in phaeozems, chernozems and umbrisols, where it decreased by 8.6–18.7% mainly due to water runoff, soil erosion, and low C input. The overall SOC stock (SOCS) in China's cropland increased by 260 Tg C (23.7 Tg C yr$^{-1}$) from 1991–2001 to 2002–2012, which was partially due to the increased crop residue return, improved fertilization and adopted conservation tillage over the period. This SOC increase represents a potential offset in carbon dioxide ($CO_2$) emissions that could help reduce the overall net $CO_2$ emissions in China.

**Keywords:** soil organic carbon; carbon stock; China's croplands; global climate change; $CO_2$ emissions

## 1. Introduction

Globally, soil organic carbon (SOC), estimated to be approximately 1500 Pg C within 1 m depth [1], is the largest organic C pool in terrestrial ecosystems, being two or three-fold greater than the C in the atmosphere or in vegetation [2,3]. As soil stores vast quantities of C, the increase in net soil C storage by even a few percent represents a substantial C sink potential [4]. Thus, enhancing SOC sequestration could be viewed as a potential approach to mitigate global warming induced by carbon dioxide ($CO_2$) emissions [5]. Soil organic carbon plays an important role in agricultural productivity [5–7]. A quantitative model was developed at the global level indicating that the yield of maize and wheat increased with increase in SOC concentrations, but yield increase levelled off when SOC concentration was approximately 2.0% [6]. The increased SOC in depleted/degraded soils could have a positive effect on crop yield [7]. They further indicated that it was difficult

to establish a direct cause–effect relationship between crop yield and SOC content due to the numerous factors including biotic and abiotic stresses. Soil organic carbon is also an important indicator of soil fertility and soil productivity, which could affect soil physical properties and biochemical processes [8,9], such as water infiltration ability, soil water retention, nutrient availability, and the biological activity of microorganisms [7,10]. Hence, understanding changes of SOC storage in cropland is useful to develop management strategies for increasing agricultural productivity and mitigating climate change.

Cropland SOC is affected by management practices (e.g., fertilizer, tillage) and environmental conditions (e.g., precipitation, temperature), which can lead to changes in soil physical and chemical properties as well impacting carbon cycling [11,12]. Fertilization is an important agricultural management practice that could directly or indirectly affect the SOC inputs and thereby change the availability of nutrients and soil carbon turnover [11,13]. Compared with traditional tillage, conservation tillage can help to reduce or prevent soil carbon losses and increase overall topsoil SOC stock (SOCS) [14–16]. Other management practices such as crop rotation, straw return and irrigation have also been shown to significantly impact SOC content [3,7,17,18]. Changes in temperatures and precipitation, wet–dry and freeze–thaw cycles affect microbial and biotic activities, which can regulate SOM decomposition and SOC accumulation [19,20]. Increases in soil temperature have been associated with higher respiration and decomposition rates, but decreases in soil moisture slow down microbial activity, which may reduce SOC decomposition [21]. Drought restricts diffusion of extracellular enzymes and soluble organic C substrates, and flooding slows oxygen diffusion to decomposition reaction sites, which all could reduce substrate availability for C decomposition [22]. In cold regions, soil C decomposition is extremely low under frozen soils in winter due to slowly diffused substrates and low activity of extracellular enzymes, but the low substrate constraints to decomposition are removed when the soil thaws in the spring [22]. Thus, management practices and environmental factors could directly or indirectly affect SOC stock by changing C input and affecting C decomposition process. Furthermore, land use changes could also be one of the important factors that affects soil carbon stock over time by influencing C input and decomposition rates [23,24].

Exploring temporal and spatial changes in SOC is importance as SOC is widely used to evaluate the impacts of past and current agricultural practices on soil quality and C cycling [17,25–28]. Globally, a number of studies on SOC change were conducted under various soil, climate and cropping systems. For example, VandenBygaart et al. [17] reported the increased rate of SOC stock ranged from 0.10 to 0.33 Mg C ha$^{-1}$ yr$^{-1}$ when considering the changes from conventional tillage to no-tillage, annual to perennial cropping and conversion of crop-fallow to continuous cropping in Canada using the Century model from 1990 to 2004. Minasny et al. [25] found that SOC increased at an average of 0.3 Tg C yr$^{-1}$ (1970–2010) for agricultural soils in South Korea, which could be partially attributed to an increase in biomass production resulting from increased fertilization and improved field management practices. Li et al. [27] demonstrated that the total SOC decreased by 7% from 1980 to 1995 and then rose by 6.2% again by 2012 in the Midwest croplands of the United States using a modelling approach. The SOC losses were primarily attributed to continuous cropping and intensive tillage practices, whereas the increases were due to the increased level of cropland production, adoption of conservation tillage and improved crop species. Zhao et al. [28] indicated that the average SOC stock in the topsoil (0–0.20 m) increased by 0.14 Mg C ha$^{-1}$ yr$^{-1}$ from 1980 to 2011 in China's croplands, which was largely attributed to increased organic inputs driven by economics and policy. Overall, the changes of global SOC stock in croplands were mainly attributed to the changes in C inputs (e.g., residue removal vs. return), fertilizer application (inorganic vs. organic fertilizer), cropping systems (monoculture vs. rotation), tillage practices (e.g., conventional vs. reduced tillage), land use change (e.g., cropland vs. forest), and climate conditions (e.g., cool vs. warm).

China has an extensive terrestrial cropland area with a diverse set of soils, cropping systems, and climatic regions, which accounts for approximately 10% of the global cultivated area [29], but it provides food for nearly 20% of the world's population [29,30]. Thus, accurate estimation of the temporal and spatial changes of SOC in China's croplands is essential to assessing the sustainability of current agricultural productivity, evaluating global $CO_2$ emissions, and examining and developing strategies to address C losses. Huang and Sun [31] and Xie et al. [32], using published data, reported an overall increasing trend of SOC in topsoil between 1980 and 2000 in China. Pan et al. [33] reported a similar trend using observations from soil monitoring studies between 1985 and 2006. However, there were some limitations in the previously conducted studies that focused on estimating spatial and temporal changes in SOC for cropland soils, including (1) the changes in SOC under different cropping systems (e.g., grain and cash crops) were ignored and (2) the insufficient field measurements or large modelling uncertainty in upscaling the results from individual sites to regional scales [34,35]. Thus, it is important to include and assess this more recent research documenting the changes in SOC density (SOCD) and stock at the national and regional scales in China over the last two decades. The International Plant Nutrition Institute (IPNI) China program has initiated a national-wide nutrient management survey in China since early 1990s and has accumulated large soil testing datasets, which provide a systematic estimation of SOC changes across the country. Therefore, the objectives of this study were (1) to evaluate the changes of SOCD for both grain and cash crops in China's croplands from 1991 to 2012; (2) to compare spatial variations of SOCD and SOCS at regional scale between the 1991–2001 and the 2002–2012; and (3) to estimate the temporal and spatial changes of SOCD under different soil types over the periods.

## 2. Materials and Methods

### 2.1. Data Source

The datasets used for analyzing SOC changes originate mainly from the International Plant Nutrition Institute (IPNI) China Program database from 1991 to 2012. The soil samplings were collected from farmers' fields, which were conducted by the IPNI Program in coordination with the local research institutes. The sampling sites were distributed across climatic zones encompassing all major agricultural regions of China (Figure 1 and Figure S1). At each site, three or four replicates of soil samples were collected and measured separately, and then the average over the replicates was calculated as the single observation. For each replicate of soil sample, three soil subsamples ($0.20 \times 0.20$ m square) were collected and mixed using a soil-sampling spade. Approximate 1 kg composite sample was obtained and placed into a ziplock bag for laboratory analysis. The SOC content was converted from soil organic matter (SOC = SOM $\times$ 0.58), which was extracted and determined using sodium hydroxide, sodium EDTA and methanol based on ASI method [36,37]. Bulk density was determined from undisturbed soil cores collected in 100 cm$^3$ cylinders after drying at 105 °C for 24 h. In total, 43,743 SOC topsoil (0–0.20 m) samples were collected and analyzed to assess the temporal and spatial variations in SOC for major agricultural regions across China's croplands. The dataset was separated into two time periods to compare the SOC changes including the 1991–2001 and the 2002–2012. There were 15,526 records (2099 sites) from the 1991–2001 and 28,217 records (3512 sites) from the 2002–2012 which included 1227 resampling sites between the two periods (Table 1). The increasing number of records in the 2002–2012 was mainly from increased sampling replicates and density in the original monitored sites. In addition, soil classification was based on World Reference Base for Soil Resources (WRB) in our study.

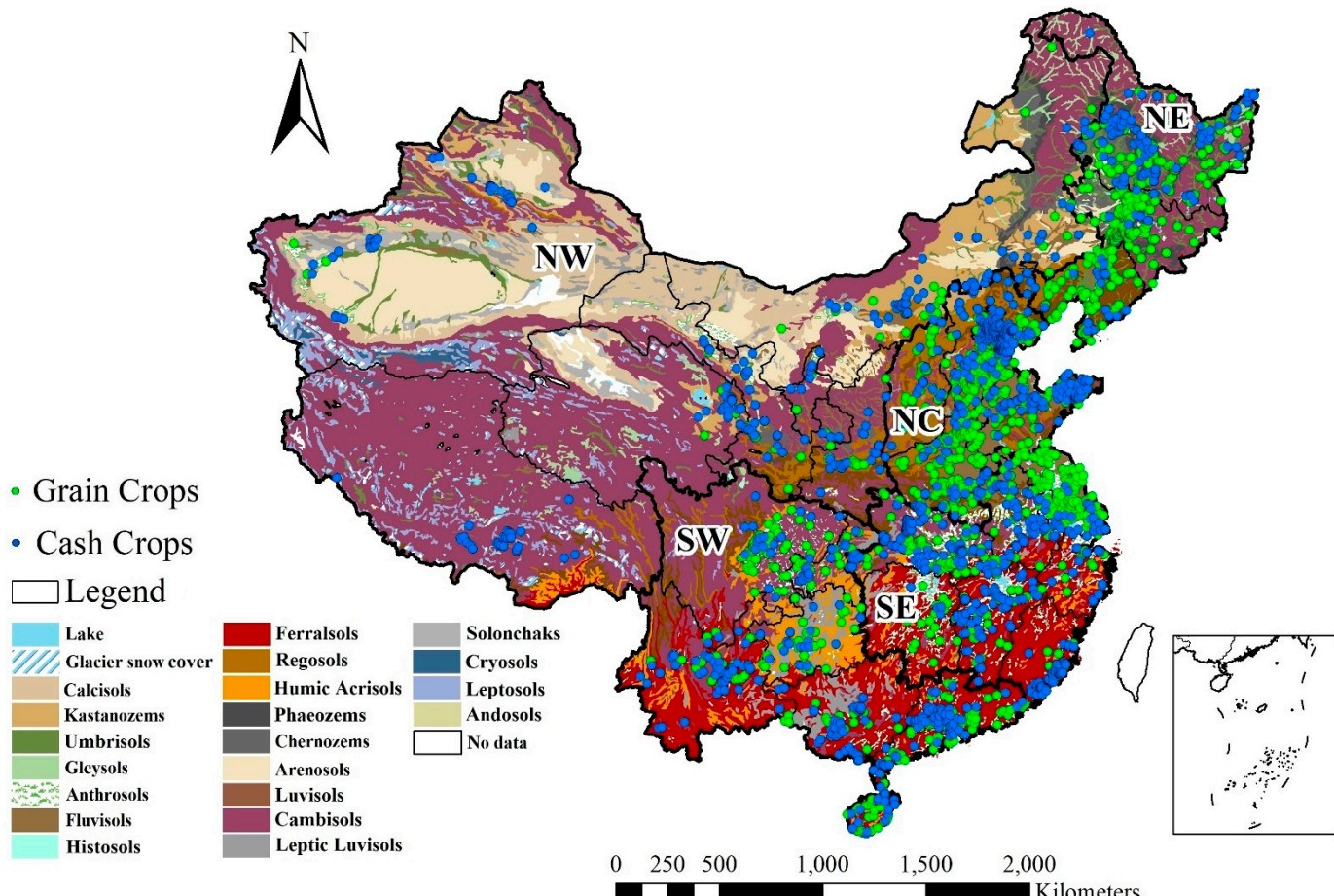

**Figure 1.** Distribution of experimental sites across five geographical regions in China's croplands from 1991 to 2012. The green and blue dots represent grain and cash crops, respectively.

**Table 1.** Numbers of experimental observation across five geographical regions and two periods of the 1991–2001 and the 2002–2012 in China's croplands.

| Item | Region [1] | Total Crops | | Grain Crops | | Cash Crops | |
|---|---|---|---|---|---|---|---|
| | | 1991–2001 | 2002–2012 | 1991–2001 | 2002–2012 | 1991–2001 | 2002–2012 |
| **Soil Test** | NE (199) [2] | 1826 | 4457 | 1344 | 3620 | 482 | 837 |
| | NC (352) | 3817 | 7095 | 2751 | 4097 | 1066 | 2998 |
| | NW (122) | 1548 | 3344 | 606 | 1201 | 942 | 2143 |
| | SE (407) | 4898 | 9785 | 3320 | 6174 | 1578 | 3611 |
| | SW (187) | 3437 | 3536 | 1851 | 2026 | 1586 | 1510 |
| | China (1227) | 15,526 | 28,217 | 9872 | 17,118 | 5654 | 11,099 |

[1] Regions used in this study are NE (Northeast), NC (North Central), NW (Northwest), SE (Southeast) and SW (Southwest). [2] The values in brackets represent the number of the resampling sites between two time periods at different regions.

Five agricultural regions were grouped based on geographical locations and China's administrative divisions to evaluate regional variation of SOC in these soils. The five regions considered in this analysis are the northeast (NE), north central (NC), northwest (NW), southeast (SE) and southwest (SW). In addition, each agricultural region was further divided into two sub-groups based on grain crops (e.g., wheat, maize, rice, soybean) and cash crops (e.g., vegetables, sunflower, rapeseed, fruit trees, cotton) [34]. The geographical samples for different agricultural regions and site information are listed in Tables 1 and 2.

**Table 2.** Summary of experimental sites from 1991 to 2012 for different geographical regions in China's croplands.

| Region [1] | Province | Main Crop | Main Soil Type (WRB Reference Soil Group) | Sample Number ($n$) | Soil pH | Precipitation (mm) | Latitude (°N) | Longitude (°E) |
|---|---|---|---|---|---|---|---|---|
| NE | Jilin, Liaoning, Heilongjiang | Maize, rice, soybean, tomato Cabbage, cucumber, flux | Phaeozems, Umbrisols, Regosols, Chernozems, Anthrosols | 6283 | 3.7–9.5 | 400–1000 | 37.74–53.53 | 118.86–135.07 |
| NC | Beijing, Tianjin, Hebei, Henan, Shandong, Shanxi | Wheat, maize, cotton, peanut Eggplant, cabbage, cucumber Tomato, cauliflower, pumpkin | Fluvisols, Regosols, Luvisols, Solonchaks | 10,912 | 3.4–10.0 | 350–900 | 31.41–42.67 | 111.25–122.63 |
| NW | Shaanxi, Ningxia, Gansu, Xinjiang, Inner Mongolia, Qinghai, Tibet | Maize, wheat, potato Cotton, carrot, cabbage Spinach, pepper, cucumber Onion, tomato, rapeseed | Cambisols, Kastanozems, Fluvisols, Calcisols | 4892 | 5.0–9.9 | 100–600 | 27.23–53.35 | 73.45–126.04 |
| SE | Hubei, Hunan, Jiangsu, Anhui, Shanghai, Jiangxi, Zhejiang, Fujian | Wheat, maize, rice Beans, cotton, sugarcane Cabbage, citrus, banana Rapeseed, sesame | Cambisols, Fluvisols, Ferralsols, Anthrosols | 14,683 | 3.6–8.8 | 700–1600 | 23.58–28.28 | 108.38–122.20 |
| SW | Chongqing, Guizhou, Yunnan, Guangxi, Hainan, Guangdong, Sichuan | Maize, wheat, rice, tea Tomato, sugarcane, rape Rapeseed, banana, cassava Pepper, pineapple | Ferralsols, Humic Acrisols, Anthrosols, Cambisols | 6973 | 3.4–8.5 | 600–2000 | 18.17–34.30 | 97.39–117.06 |

[1] Regions used in this study are NE (Northeast), NC (North Central), NW (Northwest), SE (Southeast) and SW (Southwest).

### 2.2. Soil Organic Carbon Density and Stock

The SOC density (SOCD) of a single layer of soil was calculated using the following equation:

$$\text{SOCD} = \text{SOC} \times t \times \text{BD} \times (1 - \text{Rm}) \times 10^{-1} \tag{1}$$

where SOCD is expressed in Mg C ha$^{-1}$; SOC is the C concentration in g C kg$^{-1}$; t and BD refer to the corresponding soil sampling thickness (cm) and soil bulk density (g cm$^{-3}$), respectively; and Rm is the gravimetric fraction of rock fragments (>2 mm) in soils [38,39]. In this study, the minimum value, maximum value, standard deviation and coefficient variation were used to estimate the changes of SOCD. Statistical analysis of the SOCD across geographic regions and soil types at the resampling sites between the 1991–2001 and the 2002–2012 was conducted using group T test at the 0.05 probability level ($p < 0.05$) in the SPSS 20.0 package. The missing data of topsoil bulk density in China's croplands was estimated with the topsoil SOC content using the regression equation ($R^2 = 0.7868$, $n = 410$) based on the paired data of BD and SOC content (Figure S2) from our study and the data from previous publication [40,41] as follows:

$$\text{BD} = 1.4229 \exp(-0.0062 \times \text{SOC}) \tag{2}$$

where BD represents topsoil bulk density (g cm$^{-3}$) and SOC is the SOC concentration in g C kg$^{-1}$.

The SOC stock (SOCS) of the topsoil (0–0.20 m) was calculated as follows:

$$\text{SOCS} = \text{SOCD} \times \text{Area} \tag{3}$$

where SOCS is expressed in Tg C. SOCD represents the SOC density in the 0–0.20 m depth, and Area represents cropland area (ha). The annual cropland area for grain and cash crops was used to calculate the SOC stock based on statistical data from 1991 to 2012 [30].

### 3. Results

#### 3.1. Changes of SOCD in Croplands from 1991 to 2012

It was found that the SOCD in the topsoil (0–0.20 m) generally increased across China's croplands from 1991 to 2012 for both grain and cash crops (Figure 2). There was a similar trend of increased SOCD when the results were separated into grain crops and cash crops, however, the rate of increase was a little greater for cash crops than for grain crops (Figure 2b,c). The estimated changes of SOCD based on resampling sites between the 1991–2001 and the 2002–2012 also showed an increasing trend in SOCD (Table S1 and Figure S3). The SOCD significantly increased by 7.7%, 9.0% and 5.9% from the 1991–2001 to the 2002–2012 for total crops (19.4 to 20.9 Mg C ha$^{-1}$), grain crops (18.8 to 20.5 Mg C ha$^{-1}$) and cash crops (20.3 to 21.5 Mg C ha$^{-1}$), respectively (Table S1).

#### 3.2. Temporal and Spatial Changes of SOCD and SOCS

The frequency distribution of topsoil SOCD in China's croplands in the 1991–2001 and the 2002–2012 is shown in Figure 3. The most common occurrence of SOCD was in the 10–20 Mg C ha$^{-1}$ range. For total crops, the SOCD frequency distribution in the range of 0–10 Mg C ha$^{-1}$ decreased by 41.0% from the 1991–2001 to the 2002–2012, whereas the SOCD frequency in the ranges of 10–20, 20–30 and 30–40 Mg C ha$^{-1}$ increased by 10.4%, 2.9% and 36.0%, respectively. Furthermore, a similar shift towards higher SOCD in the 2002–2012 relative to the 1991–2001 was observed for grain and cash crops. The frequency of SOCD occurring in the 0–10 Mg C ha$^{-1}$ range was decreased by 39.3% and 41.9% from the 1991–2001 to the 2002–2012 for grain and cash crops, respectively, whereas there was an increase in the higher ranges, except for the 50–60 Mg C ha$^{-1}$ range.

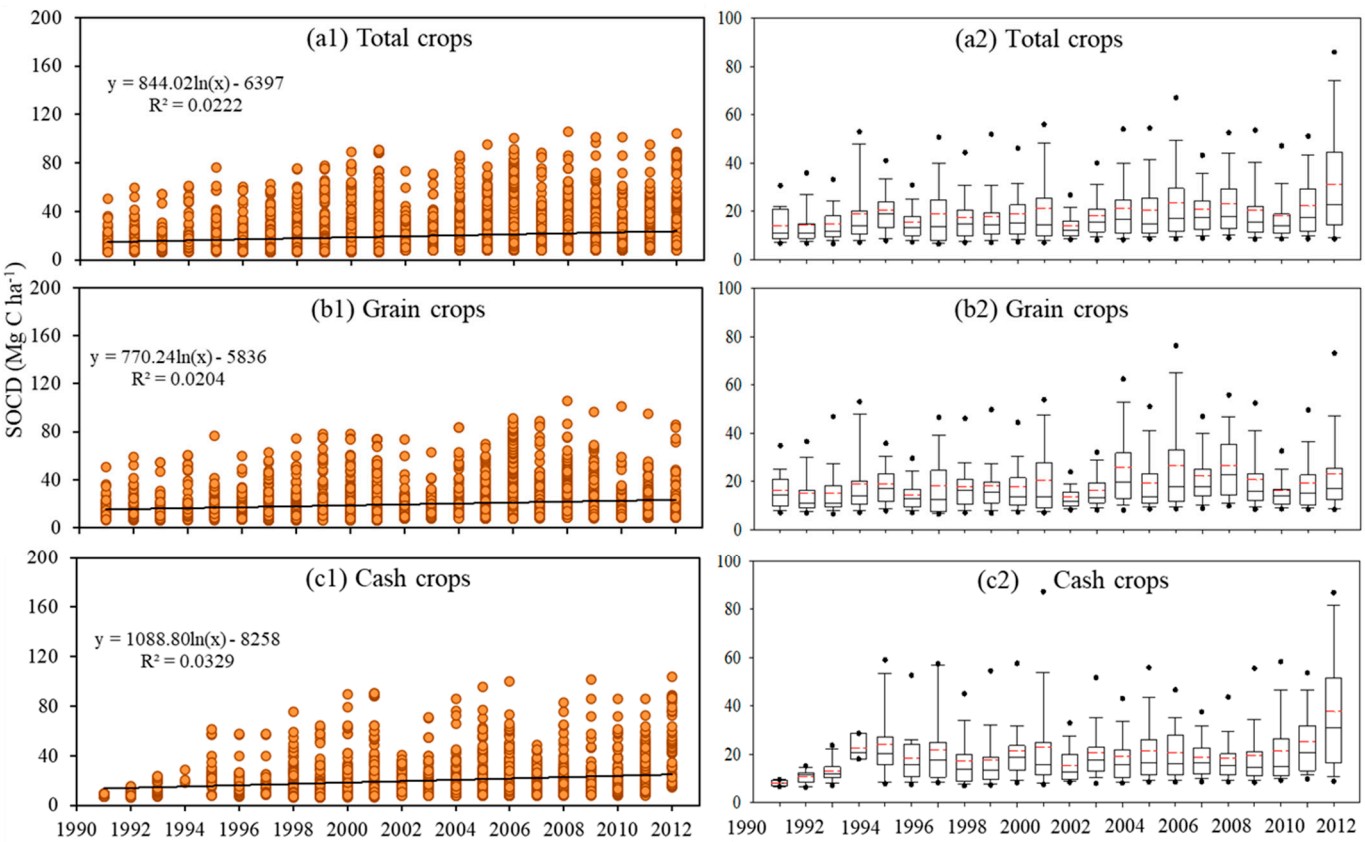

**Figure 2.** Trends of soil organic carbon density (SOCD) for (**a1**,**a2**) total crops, (**b1**,**b2**) grain crops and (**c1**,**c2**) cash crops in China's croplands from 1991 to 2012.

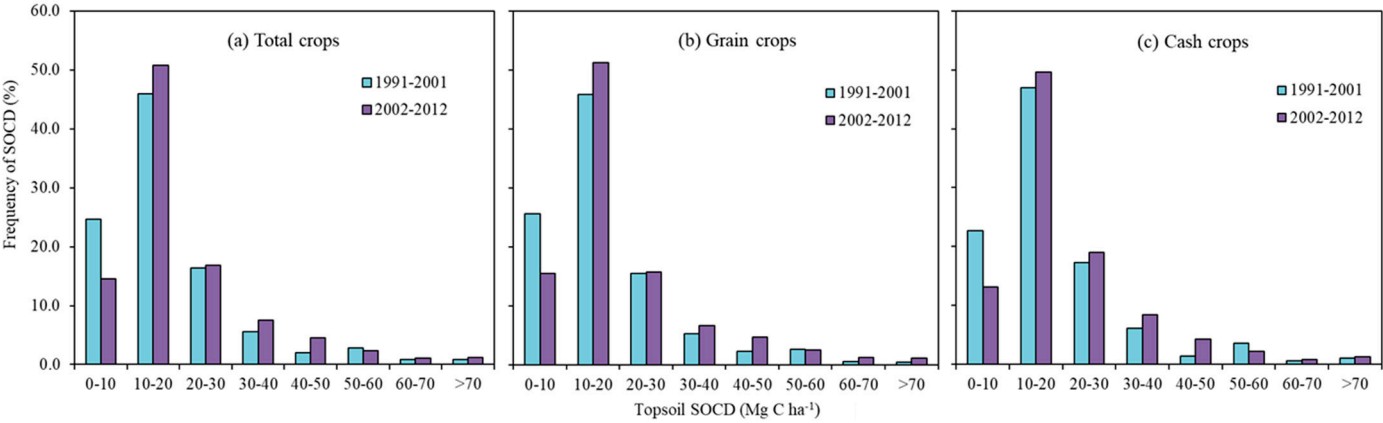

**Figure 3.** Frequency distribution of soil organic carbon density (SOCD) in the term of percentage of number of soil samples for (**a**) total crops, (**b**) grain crops and (**c**) cash crops in China's croplands between the 1991–2001 and the 2002–2012.

Estimates of the topsoil SOCD and SOCS for the entire country and for the five geographical regions were conducted to compare their variations between the 1991–2001 and 2002–2012 (Tables 3 and 4, Figure 4). For total crops, the results indicated that the topsoil SOCD in the 2002–2012 slightly decreased, relative to the 1991–2001, in the NE region (1.2%, 0.5 Mg C ha$^{-1}$), but increased in the NC (16.8%, 2.0 Mg C ha$^{-1}$), NW (17.4%, 2.2 Mg C ha$^{-1}$), SE (11.8%, 1.9 Mg C ha$^{-1}$) and SW (8.7%, 2.0 Mg C ha$^{-1}$) regions. The SOCD from resampling sites significantly increased by 16.8%, 11.8%, 12.2% and 11.8% for total crops in the NC, NW, SE and SW regions from the 1991–2002 to the 2002–2012, respectively (Table S1). The results also indicated significant increases in SOCD for both

grain and cash crops in most regions, except in the NW region for grain crops and in the SE and SW regions for cash crops. Although the SOCD decreased in the NE region, there was no significant difference between the two periods.

**Table 3.** Estimated changes of soil organic carbon density (SOCD) between the 1991–2001 and the 2002–2012 from different geographical regions in China's croplands.

| Crop Type | Item | NE [1] | | NC | | NW | | SE | | SW | | China | |
|---|---|---|---|---|---|---|---|---|---|---|---|---|---|
| | | 1991–2001 | 2002–2012 | 1991–2001 | 2002–2012 | 1991–2001 | 2002–2012 | 1991–2001 | 2002–2012 | 1991–2001 | 2002–2012 | 1991–2001 | 2002–2012 |
| Total crops | Min. [2] | 7.9 | 7.5 | 6.3 | 7.1 | 6.4 | 7.2 | 6.5 | 6.9 | 6.6 | 8.6 | 6.3 | 7.1 |
| | Median | 39.7 | 39.3 | 10.6 | 11.7 | 11.7 | 12.5 | 15.0 | 15.5 | 21.8 | 23.0 | 14.3 | 15.7 |
| | Mean | 41.1 | 40.6 | 11.7 | 13.6 | 12.7 | 14.9 | 15.8 | 17.6 | 23.4 | 25.5 | 18.6 | 20.9 |
| | Max. | 96.4 | 106.0 | 33.8 | 52.2 | 69.3 | 61.1 | 47.4 | 60.2 | 68.3 | 69.4 | 96.4 | 106.0 |
| | SD | 20.9 | 21.1 | 5.2 | 5.1 | 9.4 | 6.9 | 6.8 | 7.5 | 11.5 | 10.9 | 13.6 | 14.8 |
| | CV (%) | 51.4 | 51.2 | 42.6 | 38.9 | 63.9 | 47.1 | 40.8 | 43.3 | 48.3 | 43.7 | 73.1 | 70.8 |
| Grain crops | Min. | 7.9 | 7.5 | 6.3 | 7.1 | 6.4 | 7.2 | 6.5 | 6.9 | 6.6 | 8.6 | 6.3 | 7.1 |
| | Median | 35.6 | 36.3 | 10.5 | 11.2 | 11.7 | 12.2 | 14.9 | 14.9 | 21.4 | 20.4 | 14.0 | 15.5 |
| | Mean | 38.9 | 38.5 | 11.9 | 12.6 | 12.4 | 14.6 | 15.0 | 17.2 | 22.8 | 23.5 | 18.1 | 20.7 |
| | Max. | 80.1 | 106.0 | 33.8 | 52.2 | 53.2 | 47.1 | 47.4 | 55.1 | 68.3 | 69.4 | 80.1 | 106.0 |
| | SD | 19.2 | 20.1 | 5.2 | 4.6 | 9.5 | 6.2 | 7.1 | 6.9 | 10.7 | 10.3 | 12.9 | 14.9 |
| | CV (%) | 51.6 | 52.8 | 42.9 | 36.6 | 62.7 | 43.4 | 43.7 | 41.5 | 47.0 | 46.2 | 71.2 | 71.9 |
| Cash crops | Min. | 10.8 | 7.9 | 6.3 | 7.2 | 6.9 | 7.6 | 6.7 | 8.0 | 7.0 | 9.4 | 6.3 | 7.2 |
| | Median | 47.9 | 46.6 | 10.7 | 12.6 | 11.7 | 12.9 | 17.7 | 16.3 | 26.1 | 26.2 | 15.4 | 16.3 |
| | Mean | 49.2 | 48.9 | 11.3 | 15.1 | 13.3 | 15.1 | 17.3 | 18.3 | 25.9 | 26.9 | 19.8 | 21.2 |
| | Max. | 96.4 | 104.7 | 31.8 | 48.7 | 69.3 | 61.1 | 42.2 | 60.2 | 61.4 | 65.9 | 96.4 | 104.7 |
| | SD | 22.3 | 21.4 | 5.0 | 5.6 | 9.3 | 7.3 | 6.2 | 8.2 | 13.3 | 10.9 | 15.1 | 14.7 |
| | CV (%) | 44.1 | 42.9 | 41.5 | 40.2 | 64.7 | 48.5 | 35.3 | 44.2 | 49.0 | 40.5 | 76.1 | 69.3 |

[1] Regions used in this study are NE (Northeast), NC (North Central), NW (Northwest), SE (Southeast) and SW (Southwest). [2] Min., minimum; Max., maximum; SD, standard deviation; CV, coefficient variation.

**Table 4.** Estimated changes of soil organic carbon stock (SOCS) between the 1991–2001 and the 2002–2012 from different geographical regions in China's croplands.

| Crop Type | Region [1] | Crop Area [2] | SOCS (Tg C) | | Total Changes | | Annual Changes |
|---|---|---|---|---|---|---|---|
| | | (Mha) | 1991–2001 | 2002–2012 | (Tg C) | (%) | (Tg C yr$^{-1}$) |
| Total crops | NE | 18.5 | 759 | 749 | −9.8 | −1.3 | −0.9 |
| | NC | 37.7 | 439 | 513 | 73.9 | 16.8 | 6.7 |
| | NW | 19.7 | 250 | 293 | 43.3 | 17.4 | 3.9 |
| | SE | 43.8 | 691 | 772 | 81.6 | 11.8 | 7.4 |
| | SW | 35.1 | 823 | 894 | 71.3 | 8.7 | 6.5 |
| Grain crops | NE | 16.0 | 622 | 617 | −5.7 | −0.9 | −0.5 |
| | NC | 27.0 | 320 | 339 | 19.2 | 6.0 | 1.7 |
| | NW | 13.9 | 173 | 203 | 29.4 | 17.0 | 2.7 |
| | SE | 28.3 | 426 | 488 | 61.7 | 14.5 | 5.6 |
| | SW | 23.5 | 535 | 551 | 16.0 | 3.0 | 1.5 |
| Cash crops | NE | 2.5 | 121 | 120 | −0.9 | −0.7 | −0.1 |
| | NC | 10.7 | 121 | 162 | 40.8 | 33.7 | 3.7 |
| | NW | 5.8 | 77 | 87 | 10.4 | 13.6 | 0.9 |
| | SE | 15.5 | 268 | 282 | 14.1 | 5.3 | 1.3 |
| | SW | 11.6 | 301 | 313 | 12.0 | 4.0 | 1.1 |
| Topsoil | China | 154.7 | 2961 | 3221 | 260 | 8.8 | 23.7 |

[1] Regions used in this study are NE (Northeast), NC (North Central), NW (Northwest), SE (Southeast) and SW (Southwest). [2] The values were collected from the China Statistical Yearbook (1991–2012) [30].

In this study, the topsoil SOCS increased by 260 Tg C (23.7 Tg C yr$^{-1}$) from the 1991–2001 to the 2002–2012 across all the sampling regions (Table 4). For total crops, the annual changes of SOCS were −0.9, 6.7, 3.9 7.4 and 6.5 Tg C yr$^{-1}$ between the 1991–2001 and the 2002–2012 in the NE, NC, NW, SE and SW regions, respectively. In addition, there were significant differences in the temporal and spatial changes of SOCS between grain crops

and cash crops from the 1991–2001 to the 2002–2012. For grain crops, compared with the 1991–2001, SOCS in the 2002–2012 increased by 6.0%, 17.0%, 14.5% and 3.0% in the NC, NW, SE and SW regions, respectively. However, SOCS decreased by 0.9% in the NE. The SOCS for cash crops in the 2002–2012 increased by 33.7%, 13.6%, 5.3%, 4.0% in the NC, NW, SE and SW regions, respectively, compared to those in the 1991–2001.

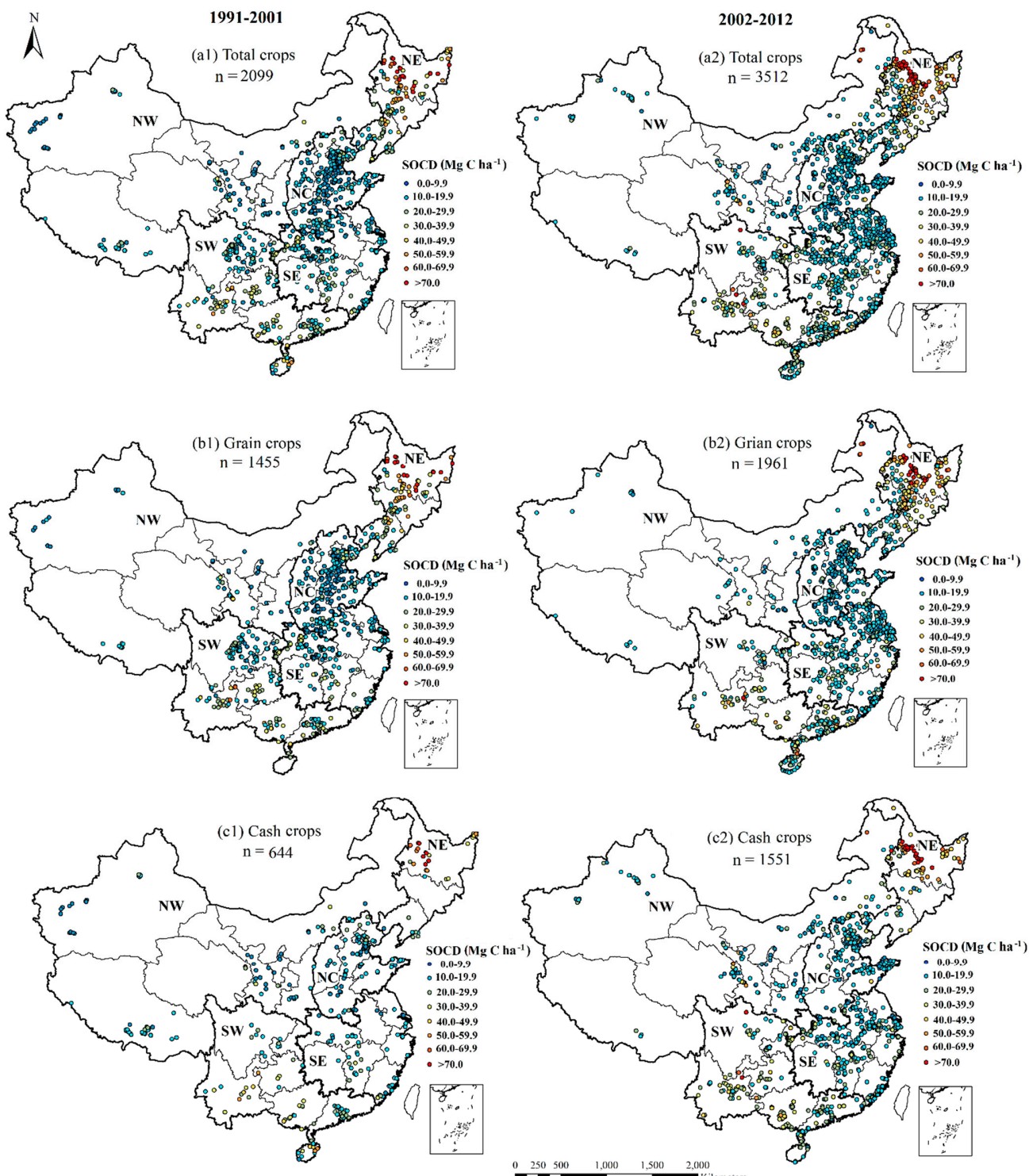

**Figure 4.** Geographical variation of soil organic carbon density (SOCD) for (**a1,a2**) total crops, (**b1,b2**) grain crops and (**c1,c2**) cash crops in China's croplands between 1991–2001 and 2002–2012.

### 3.3. Changes of SOCD under Different Soil Types

At the regional scale, the magnitude of the SOCD changes varied greatly across the main soil types (Figure 5, Table S2). For total crops, compared with the 1991–2001, the SOCD in the 2002–2012 increased by 37.0 and 10.2% in anthrosols and regosols in the NE, 18.5, 28.3 and 19.1% in fluvisols, regosols and luvisols in the NC, 17.2, 17.3 and 30.4% in cambisols, fluvisols and calcisols in the NW, 23.1, 20.1 and 28.0% in cambisols, fluvisols and ferralsols in the SE, and 9.3, 28.4, 6.8 and 44.2% in ferralsols, humic acrisols, anthrosols and cambisols in the SW region, respectively. Nevertheless, the SOCD decreased by 8.6, 18.7 and 8.9% in phaeozems, umbrisols and chernozems in the NE region, 2.9% in kastanozems in the NW region and 0.9% in anthrosols in the SE region. The same trend of SOCD in grain and cash crops were observed among most of the soil types, but a few reverse trends were found. For example, the SOCD in ferralsols increased in grain crops, but decreased in cash crops in the SW. At the resampling sites, the SOCD significantly increased by 13.1%, 25.4% and 17.2% for total crops, 12.1%, 28.3% and 21.0% for grain crops, and 14.5%, 23.1% and 14.4% for cash crops in fluvisols, regosols and luvisols in the NC region from the 1991–2001 to the 2002–2012, respectively. In addition, there were significant increases in SOCD for total crops in cambisols (21.1%), fluvisols (10.7%) and ferralsols (18.3%) in the SE region and in cambisols (21.5%) in the NW region from the 1991–2001 to the 2002–2012. For grain crops, the SOCD significantly increased by 25.4%,16.0% and 26.5% in cambisols, fluvisols and ferralsols, respectively, in the SE region and 20.5% in cambisols in the SW region between the two periods (Figure S4).

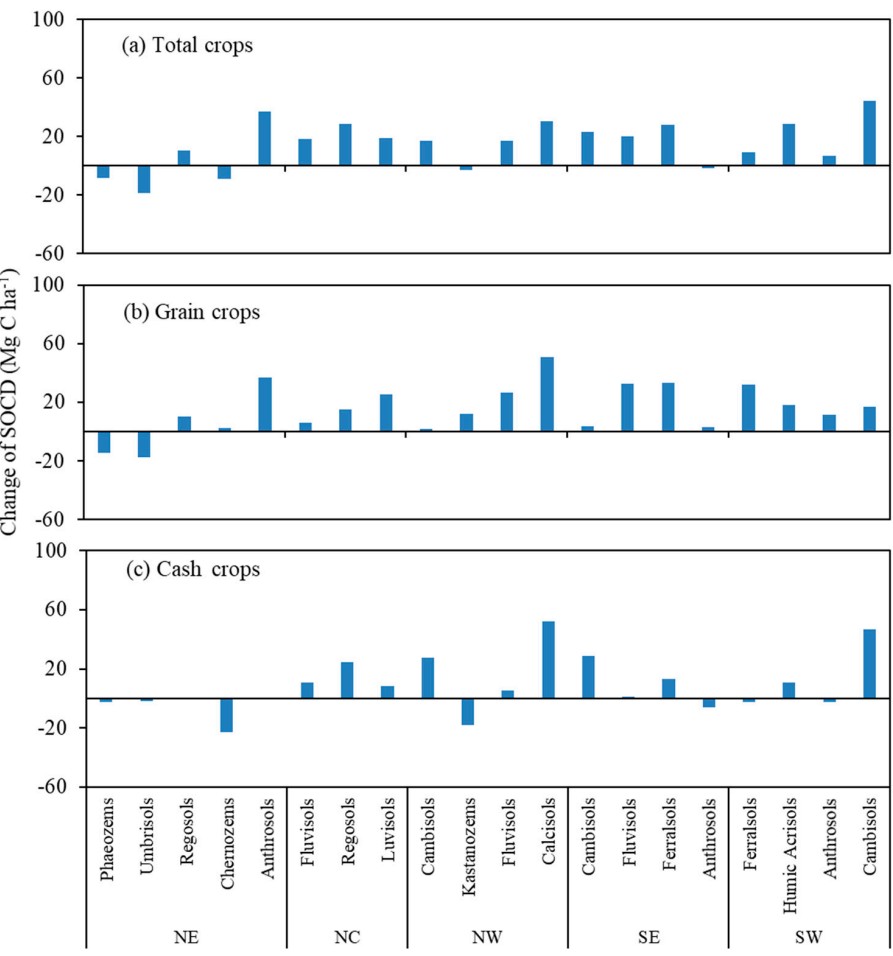

**Figure 5.** Changes of soil organic carbon density (SOCD) under different soil types for (**a**) total crops, (**b**) grain crops and (**c**) cash crops in China's croplands between the 1991–2001 and the 2002–2012. NE (Northeast), NC (North Central), NW (Northwest), SE (Southeast) and SW (Southwest).

## 4. Discussion

### 4.1. Changes of SOCD and SOCS at National Scale

The increasing trend in SOC, as determined in our study, which might be caused by higher rates of increased fertilizer application and crop yields with greater rates of increased crop C inputs in the 1991–2001 relative to the 2002–2012 [42] (Table 5). The overall trend of increased SOCD in China might be caused by the improved farm management practices. These included adoption of balanced fertilization, increased organic fertilizer (human and livestock waste) inputs and higher amounts of crop residues returned to soils for different geographical regions in China's croplands between the 1991–2001 and the 2002–2012. Similar to other countries [17,43], some field management implemented by farmers in China have contributed to SOC increase, including the increase of organic fertilizer input [44,45], improvement in crop cultivars [46] and conversion of conventional tillage to minimum or no tillage [15,47]. Previous studies indicated that the SOC stock was positively correlated with crop residue C, N fertilizer and organic fertilizer inputs [18,28,48]. The SOC sequestration and $CO_2$ emission reduction could be achieved by optimizing management practices (e.g., residue return combined with NPK fertilizer, organic manure application, and mixed application of organic and inorganic fertilizers) [49,50]. It has been reported that the SOC stock would increase by $\geq$ 25.0 Tg C yr$^{-1}$ or 0.63% yr$^{-1}$ through the implementation of the recommended management practices in China's croplands, compensating for $CO_2$ emissions by 1.0% [51]. In addition, Zhao et al. [28] further indicated that application of N fertilizer increased SOC at low SOC cropland, which was attributed to the increased crop dry matter production, resulting in more C inputs to soils. However, excessive chemical N fertilizer may constrain SOC sequestration through stimulating decomposition of cellulose-dominant crop residues, reduce C retention efficiency by favoring bacteria over fungi, and even limit root growth by stimulating soil acidification [28].

**Table 5.** Changes in chemical and organic fertilizer inputs, straw returned to soils and crop production for different geographical regions in China's croplands between the 1991–2001 and the 2002–2012.

| Region [1] | Chemical Fertilizer [2] (Tg) | | Organic Manure [3] (Tg) | | Straw Returned [4] (Tg) | | Total Crop Production [5] (Tg) | |
|---|---|---|---|---|---|---|---|---|
| | 1991–2001 | 2002–2012 | 1991–2001 | 2002–2012 | 1991–2001 | 2002–2012 | 1991–2001 | 2002–2012 |
| NE | 3.2 * | 4.7 | 174.7 | 254.9 | 7.3 | 13.4 | 103.9 | 142.3 |
| NC | 10.5 | 14.9 | 549.7 | 672.7 | 63.8 | 102.7 | 274.2 | 441.5 |
| NW | 3.2 | 5.7 | 230.2 | 268.6 | 5.7 | 10.5 | 76.4 | 138.7 |
| SE | 12.3 | 15.3 | 537.5 | 701.0 | 33.5 | 45.3 | 287.0 | 368.7 |
| SW | 7.4 | 10.4 | 598.9 | 747.9 | 25.5 | 34.6 | 248.3 | 353.0 |

[1] Regions used in this study are NE (Northeast), NC (North Central), NW (Northwest), SE (Southeast) and SW (Southwest). [2] Chemical fertilizer indicates total N, P, and K nutrients from inorganic fertilizers. [3] Organic manure indicates total amount of human and livestock waste applied to soils. [4] Straw returned indicates the amount of crop residues returned to soils. [5] Total crop production indicates total production of grain and cash crops. * The values were collected from the China Statistical Yearbook (1991–2012) [30,42].

Our results overall fell into the range of SOCS changes (9.6–27.6 Tg C yr$^{-1}$) in topsoil in China's croplands reported in previous studies (Table S3). Variations between studies were mainly due to different data source, monitoring periods, cultivation history, management practices and cropland area [28,31,32,52–54]. For example, Xie et al. [32] reported that the SOC stocks in China's croplands (156 Mha) increased by 472 Tg C with an average rate of 23.6 Tg C yr$^{-1}$ from the 1980s to the 2000s based on the Second National Soil Survey and published data. Zhao et al. [28] indicated that the SOC stock in topsoil increased by 560 Tg C with a rate of 18.1 Tg C yr$^{-1}$ in topsoil croplands (130 Mha) from 1980–2011. Comparatively, the SOCS increase rate was 19.9 Tg C yr$^{-1}$ in this study when we convert the area to 130 Mha, which was comparable to the SOCS increase rate in previous studies (Table S3). A previous study reported that SOC in the topsoil of farmland in China (no data for Hainan, Tibet and Yunnan) increased in 53–59%, decreased in 30–31% and stabilized in 4–6% from the early 1980s to the late 1990s [31]. Yu et al. [52], in a study using the

Agro-C model, revealed that SOC stock (0–0.30 m) ranged from 11.0–36.5 Tg C yr$^{-1}$ and increased in approximately 81% of Chinese croplands from 1980 to 2009, which was due to the increased crop production and retention of crop residues. However, a study using the DNDC (DeNitrification-DeComposition) model simulated decreasing trends in SOC stock with annual losses of 1.6–2.0% [55,56]. This was mainly a result of underestimated crop residues being returned to the soil over time. In addition, the changes of SOCD in this study were closely related to the variations of crop yields for grain and cash crops across five regions during the same period (1991–2012) [34]. Overall, the increased SOC sequestration in China's croplands is identified over two decades as the result of the increased organic C input with greater returned crop residue, appropriate inorganic and organic fertilizer application and improved conservation tillage [28,51]. The net increase of SOCS in China's croplands would help contribute to offsetting $CO_2$ emissions and be an important factor in mitigating potential climate change impacts during the 2002–2012. Furthermore, previous reports indicated that there were great potentials for C sequestration under improved agricultural management strategies in China's croplands [39,51]. The SOC will increase asymptotically with additional C input and reach an equilibrium C level, especially for low SOC soil over a certain time, but the rate of increase of equilibrium SOC declines gradually without continuous C input indefinitely [57]. The increasing C input no longer results in additional soil carbon when SOC at steady level reaches the ceiling [39,57]. Thus, it is worth considering that the less of C budget would remain for future exploitation by improving management practices, especially for high SOC soils. These improved management practices efforts to increase SOC sequestration and to restore productivity should be further concentrated on the sites with lower SOC.

### 4.2. Changes of SOCD and SOCS at Regional Scale

Similar trends of the SOCD were observed for grain crops and cash crops in different regions between 1991–2001 and 2002–2012. The increased SOCD was detected across all regions in China's croplands, except for Northeast China (approximately a 0.5 Mg C ha$^{-1}$ of SOC loss), which was consistent with a previous study, which reported that the about 0.41 Mg C ha$^{-1}$ decreased in the NE region [28]. The slight decrease in SOCD in the NE region was partially due to the negative impacts of the relative low straw return rate, high initial SOC, soil erosion and degradation as well as climate change (e.g., drought, temperature stress) [28,58,59]. In addition, different agricultural management practices in each region, such as the use of chemical fertilizer, manure and the retention of crop straw (Table 5), would affect the distribution of SOCD. Statistical data could indicate the changes trend of nutrient input from chemical fertilizer, manure and straw return at the regional scale. Although the resolution of nutrient input data from statistical data did not perfectly match measurement data from actual field in our study, both datasets were collected based on farmers' croplands. Thus, regional nutrient input data was a supplement of actual field measurements and could reflect the changes of famers' field practices over time which provided an explanation for regional SOC changes. In addition, regional climate change impacts can be beneficial or harmful to agricultural systems depending on local climatic conditions and management practices. For example, increasing precipitation contributes to an increase in SOCD in northern China from the arid to semi-humid zone due to the increase in vegetation productivity and the increase of SOCD from tropical to cold-temperate zone in the eastern part of the country, which was likely attributed to decreased mineralization caused by decreased temperature from the south to the north [41]. Moreover, the changes in the spatial pattern of SOC can be attributed partly to the effects of soil types and texture, which have significant impacts on the soil water holding capacity and the decomposition rate of SOC [60]. Furthermore, the increased SOC stock was observed in our study across most agricultural regions in China's croplands, indicating that the existing SOC may still has a potential to increase mainly due to that the SOC stock has not reached the upper limit of the saturation in these regions [39]. However, the decreased SOC stock in the NE region was detected, possibly because the new and lower equilibriums have not been

reached yet due to a relative short cultivation history after conversion from forests and grasslands with high initial SOC [61,62]. It may reflect a great challenge of how to alleviate soil degradation in this region and achieve the C sequestration and emission reduction through effectively improved field management practices. Although $CO_2$ leakage from carbon capture and storage was not analyzed in our study, recent study has shown that radial diffusion of $CO_2$ leakage caused farmland degradation, degraded the morphological characteristics of spring wheat, and the impacts of $CO_2$ leakage on horizontal topsoil $CO_2$ concentrations was 2.67 times that of the source depth [63]. The SOC saturation, reversibility and leakage risk under different SOC levels at the regional scale should be considered in the future investigations.

### 4.3. Changes of SOCD under Different Soil Types

In our study, the decreased SOCD in phaeozems in the NE region from the 1991–2001 to the 2002–2012 was consistent with Xie et al. [32] and Wang et al. [64], which was partially attributed to the soil erosion by intense tillage in phaeozems region. Tillage erosion has been reported as an important erosion process contributing to SOC redistribution, which could result in SOC depletion, especially in the top and lower section of slopes [65]. Reducing tillage depth and size of implements, adopting tillage direction to upslope–downslope tillage, and contouring tillage patterns should be considered as effective strategies to reduce the negative impacts on tillage erosion [65]. Huang and Sun [31] reported the significant decrease of SOC in phaeozems and umbrisols in the NE region and Yan et al. [61] indicated the decreased SOCD in phaeozems in this region mainly due to water runoff, soil erosion and low C input. In addition, Wang et al. [64] indicated that the SOC loss in phaeozems region mainly derived from the decreases of slow and resistant SOC fractions. The increase of SOCD in anthrosols was observed in the NE, being attributed to the greater dry matter production and C sequestration efficiency than dry upland [66,67]. Increases of SOCD in anthrosols were also found in the SE and SW regions for grain crops; however, the SOCD decreased in anthrosols for cash crops because of the increased C mineralization by improving soil aeration [68]. The increased SOCD in fluvisols was found in the NC region in this study, which was consistent with Huang and Sun [31] and Xie et al. [32]. High rates of straw returned for major crops to cropland [69,70], application of combined organic manure and inorganic fertilizer [71], and development of no-tillage and reduced-tillage methods [15] helped contribute to the higher overall crop production, which resulted in increased root exudates and crop C inputs to soil and ultimately to higher SOC contents. The SOCD increase in the SE and SW regions, especially for grain crops, where ferralsols are mainly distributed, was associated with its high C saturation deficit due to the low initial SOC level [72] and organic fertilizer application [73,74].

### 4.4. Uncertainty and Limitations

Although we compared the SOC changes under different soil and crop types over two decades using large datasets, the uncertainties in estimating regional changes of SOC could be caused by the lack of more paired resampling data over time, detailed field management practices (e.g., fertilizer, crop residue, tillage, and irrigation) and precise cultivation history. The change of land use patterns in the past decades with the rapid development of urbanization and economic could also result in the uncertainty of resampling in the same site over time. The same method/procedure was used to reduce the deviation caused by the laboratory analysis errors, but huge soil samples were tested in different cooperating institutes under the IPNI program which could lead to uncertainties. In addition, the unavailable BD values were calculated by the regression equation based on the measurements in this study and the collected data from previous publications from 1991 to 2012. However, the changes of BD values could be affected by the different land use patterns, cultivation history, vegetation, and soil types, which may lead to the underestimation or overestimation. Furthermore, there is no accurate data in the changes of cultivated land area, especially for area changes of grain crops and cash crops every year, thus the

SOCD would be affected by the new increased or reduced cropland area. It is important to implement more long-term monitoring studies to improve our understanding of the impacts of management practices, climate, and soils on SOC changes in China.

## 5. Conclusions

Spatial and temporal variation in topsoil organic carbon dynamics was estimated based on large soil testing datasets in China's croplands. The increased SOC sequestration was identified between the 1991–2001 and 2002–2012 periods as a result of the increased organic C input with greater returned crop residue, appropriate inorganic and organic fertilizer application and improved conservation tillage. Spatial variation of topsoil SOC differed significantly among five agricultural regions mainly due to the differences in climate conditions, land use and soil properties. Additionally, the SOC increased in most major soil types across regions in China's croplands, but it decreased in phaeozems, chernozems and umbrisols being attributed to water runoff, soil erosion and low C input. Since SOC levels in soils are known to be highly variable spatially, the extensive array of soil monitoring stations used in this study were valuable for understanding and encapsulating the differences between climatic zones, soil types and crops grown. In regions that were identified as having decreased SOC levels and degraded soils, alternative management practices could be implemented to combat this trend in SOC loss and soil degradation. Although we compared the changes in SOC under different soil and crop types over two decades using large datasets, the lack of more paired resampling data over time, detailed field management practices (e.g., fertilizer, crop residue, tillage and irrigation) and precise cultivation history led to the uncertainties in estimating the regional changes of SOC. In order to better understand soil carbon cycling and to identify management practices that benefit C sequestration, it is important that combination of process-based models and long-term monitoring sites are more effective to estimate the response of SOC stock to different soil types and cropping systems under climate variability. Additionally, future research should focus on exploring the effects of various field management practices with potential to build C in deeper soil depths.

**Supplementary Materials:** The following are available online at https://www.mdpi.com/article/10.3390/agronomy11071433/s1, Figure S1: Number of soil samples from 1991 to 2012 in China's croplands. Figure S2: Correlation of bulk density with soil organic carbon (0–0.20 m depth) in China's croplands. Figure S3. Soil organic carbon density (SOCD) for (a) total crops, (b) grain crops and (c) cash crops for different geographical regions in China's croplands between the 1991–2001 and the 2002–2012. NE (Northeast), NC (North Central), NW (Northwest), SE (Southeast) and SW (Southwest). Figure S4. Changes of soil organic carbon density (SOCD) for resampling sites under different soil types for (a) total crops, (b) grain crops and (c) cash crops in China's croplands between the 1991–2001 and the 2002–2012. NE (Northeast), NC (North Central), NW (Northwest), SE (Southeast) and SW (Southwest). Table S1: Estimated changes of soil organic carbon density (SOCD) for resampling sites between the 1991–2001 and the 2002–2012 from different geographical regions in China's croplands. Table S2: Estimated changes of soil organic carbon density (SOCD) under different soil types between the 1991–2001 and the 2002–2012 from different geographical regions in China's croplands. Table S3: Comparisons of the estimated topsoil soil organic carbon stock (SOCS) in China's croplands.

**Author Contributions:** W.H., conceptualization, methodology, formal analysis, writing—original draft preparation; P.H., conceptualization, resources, supervision, project administration and funding acquisition, writing—review and editing; R.J., conceptualization, methodology, validation, writing—original draft; J.Y. and C.F.D., methodology and writing—review and editing; W.N.S. and B.B.G., methodology, software, writing—review and editing; W.Z., conceptualization, resources and writing—review and editing. All authors have read and agreed to the published version of the manuscript.

**Funding:** This research was funded by National Natural Science Foundation of China (No. 31972515), China Agricultural Research System (CARS-09-P31), and the International Plant Nutrition Institute China Program.

**Institutional Review Board Statement:** Not applicable.

**Informed Consent Statement:** Not applicable.

**Data Availability Statement:** Relevant data applicable to this research are within the paper.

**Acknowledgments:** We thank the cooperators from 31 provinces in China for taking soil samples and managing the field experiments.

**Conflicts of Interest:** The authors declare no conflict of interest.

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
