# Peer review of "Soil Organic Carbon Changes for Croplands across China from 1991 to 2012"

_agronomy, doi:10.3390/agronomy11071433_

Round 1
Reviewer 1 Report
Review
Abstract
1.Lines 20-22. A frequency distribution 20 (%) of SOCD in 10 Mg C ha-1 ranges comparing the changes from the 1991–2001 to the 2002–2012 21 indicated that the SOCD increased for all intervals except in the 0–10 and 50–60 Mg C ha-1 intervals 22 for both grain and cash crops.
Minor grammar inconsistencies. This sentence could be improved for purposes of better communication of the science.
- Line 25-28. The overall SOC stock for agricultural 25 soils in China increased by 260 Tg C (23.7 Tg C yr−1) from the 1991–2001 to the 2002–2012, which 26 corresponds to that crop residue return, improved fertilization and conservation tillage became 27 more widespread over a time period in China.
Minor grammar inconsistencies. This sentence could be improved for purposes of better communication of the science.
- The abstract could briefly mention the soil database that this work is based on. These >43K samples were not collected just for the purposes of this study. Further, it’s not clear from the abstract text how increases on SOC were attributed to crop return, improved fertilization and conservation tillage. Were these land managements recorded as part of the large overarching project. This aspect could be clearer.
Introduction
- Overall, this section is written well. However, the aims of the study are simply stated as being to evaluate the spatio-temporal variation of SOC. The specific hypotheses could be more clearly defined here.
For example, the dataset was split into two time periods, 1991-2001 and 2002-12. This could be rationalised based on the perspective of a study aim.
Materials and Methods
- In the main the methods section is adequate. The maps and featured tables are detailed, providing a constructive addition to the manuscript. Is the IPNI dataset open access? Further, what is meant by the ‘cooperating institutes’. Could this be explained or defined further? The author team have published a number of papers from the IPNI dataset (see below), could these perhaps be summarised in the SI?
Temporal and spatial changes in soil available phosphorus in China (1990-2012) Field Crops Research. Volume 192, Pages 13 – 201 June 2016. Doi. 10.1016/j.fcr.2016.04.006
- Bulk density is a critical factor, but how this was determined, either field or laboratory measurements is not stated. This needs to be addressed, the method justified, and any shortcoming discussed.
Results
- The companion land management and nutrient input data has been sourced from the China Agriculture Yearbook (1992-2012. How this data matches the resolution of the actual field soils collected, could be discussed in more depth.
- Line 241. Grammar error. Correction needed.
- Line 242. Here and throughout the text the way the two temporal categories are referred to can be improved for the purposes of readability. For example, “the 1991–2001 and the 2002–2012 varied”.
- Line 269. Rephrase. “had occurrences of 53–59% increase”.
- Define the DNDC model in full.
- Line 286-289. The slight decrease mentioned doesn’t appear so insignificant when compared to the other regions where on average SOCD gains were made. The data for the NE should also placed in additional context taking into account the increase in SOCD for other regions.
- Line 219-292. The positive correlation was found between the SOC stock and N fertilizer and crop residue C input [28]. Minor grammar inconsistencies. Sentence could be improved for purposes of better communication of the science.
- Line 306. “that an increase of 472 Tg in C pool with the accumulative rate of 23.6 Tg yr-1”. Sentence could be improved for purposes of better communication of the science.
- Line 319-321. In fact, the more of the potential carbon storage budget that has been achieved already historically, the less of this budget that remains for future exploitation by improving management practices. This is an important point but could be articulated more clearly.
- Line 351. precise cultivation history led to the uncertainties in estimating regional changes of SOC. Minor grammar inconsistencies. Sentence could be improved for purposes of better communication of the science.
- Line 355. laboratory analysis errors, but the huge soil samples were tested in different cooperating. Minor grammar inconsistencies. Sentence could be improved for purposes of better communication of the science.
Summary
An important topic that would interest many, both within China and beyond. An extensive dataset with excellent spatial and temporal coverage. The findings are of high interest, although, they also largely support the trends in other works, the scale and scope of this study provide additional value.
Not all the relevant methods are described.
There could be more discussion on how the reported trends relate to sink saturation, reversibility risk and leakage risk.
One of the main conclusion messages was that more long-term field trials are needed. However, really how feasible is this? Do we have the luxury to wait for another 20 years for sufficient new data to be generated? What about advances in measurements that provide quicker indications of improved carbon capture? or better harnessing of existing field sites? This could be discussed.
Also, the point about harnessing or building C in non-top soils could be explained in more depth. Does this refer to extending woodlands or hedgerows? Or trying to enhance C storage in deeper soil horizons.
The manuscript still requires further proofreading and editing. My comments above are not exhaustive.
Overall though I consider this work suitable for publication pending some further revision.
Author Response
Abstract
Point 1: Lines 20-22. A frequency distribution 20 (%) of SOCD in 10 Mg C ha-1 ranges comparing the changes from the 1991–2001 to the 2002–2012 21 indicated that the SOCD increased for all intervals except in the 0–10 and 50–60 Mg C ha-1 intervals 22 for both grain and cash crops.
Minor grammar inconsistencies. This sentence could be improved for purposes of better communication of the science.
Response 1: We have revised the sentence as follow (Lines 22-25):
“For both grain and cash crops, the SOCD frequency distribution (%) increased in the ranges of 10–20, 20–30 and 30–40 Mg C ha−1 from the 1991-2001 to the 2002-2012 but decreased in the ranges of 0–10 and 50–60 Mg C ha−1.”
Point 2: Line 25-28. The overall SOC stock for agricultural 25 soils in China increased by 260 Tg C (23.7 Tg C yr−1) from the 1991–2001 to the 2002–2012, which 26 corresponds to that crop residue return, improved fertilization and conservation tillage became 27 more widespread over a time period in China.
Minor grammar inconsistencies. This sentence could be improved for purposes of better communication of the science.
Response 2: We revised the sentence as follow (Lines 29-32):
“The overall SOC stock in China’s cropland increased by 260 Tg C (23.7 Tg C yr−1) from the 1991–2001 to the 2002–2012, which was partially due to the increased crop residue return, improved fertilization and adopted conservation tillage over the period.”
Point 3: The abstract could briefly mention the soil database that this work is based on. These >43K samples were not collected just for the purposes of this study. Further, it’s not clear from the abstract text how increases on SOC were attributed to crop return, improved fertilization and conservation tillage. Were these land managements recorded as part of the large overarching project. This aspect could be clearer.
Response 3: The soil database in our study was collected from the International Plant Nutrition Institute (IPNI) China Program, which was a part of the large overarching project. We have added the details program of the soil database in abstract in the new manuscript (Lines 17-18).
We revised the sentence (Lines 22-25) to explain that the increased SOC could be partially due to the increased crop residue return, improved fertilization and adopted conservation tillage over the period, based on the statistical data.
Introduction
Point 4: Overall, this section is written well. However, the aims of the study are simply stated as being to evaluate the spatio-temporal variation of SOC. The specific hypotheses could be more clearly defined here. For example, the dataset was split into two time periods, 1991-2001 and 2002-12. This could be rationalised based on the perspective of a study aim.
Response 4: We revised the objectives of this study as follows (Lines 120-126):
“Therefore, the objectives of this study were (1) to evaluate the changes of SOCD for both grain and cash crops in China’s croplands from 1991 to 2012; (2) to compare spatial variations of SOCD and SOCS at regional scale between the 1991-2001 and the 2002-2012; and (3) to estimate the temporal and spatial changes of SOCD under different soil types over the periods.”
Materials and Methods
Point 5: In the main the methods section is adequate. The maps and featured tables are detailed, providing a constructive addition to the manuscript. Is the IPNI dataset open access? Further, what is meant by the ‘cooperating institutes’. Could this be explained or defined further? The author team have published a number of papers from the IPNI dataset (see below), could these perhaps be summarised in the SI?
Temporal and spatial changes in soil available phosphorus in China (1990-2012) Field Crops Research. Volume 192, Pages 13 – 201 June 2016. Doi. 10.1016/j.fcr.2016.04.006
Response 5: Currently, the IPNI dataset is not open access mainly due to the fact that the dataset was collected by different institutes. Cooperative institution refers to the research institutes that cooperated with the IPNI program. The revised sentence showed as follows (Lines 131-133):
“The soil samplings were collected from farmers’ fields, which were conducted by the IPNI program in coordination with the local research institutes.”
Additionally, the IPNI dataset has been used to analyze the temporal and spatial changes in soil available potassium and phosphorus with two published papers (He et al., 2015; Ma et al., 2016), which were cited in our study.
References:
He, P., Yang, L.P., Xu, X.P., Zhao, S.C., Chen, F., Li, S.T., Tu, S.H., Jin, J.Y., Johnston, A., 2015. Temporal and spatial variation of soil available potassium in China (1990–2012). Field Crops Research, 173, 49-56.
Ma, J., He, P., Xu, X., He, W., Liu, Y., Yang, F., Chen, F., Li, S., Tu, S., Jin, J., Johnston, A.M., Zhou, W., 2016. Temporal and spatial changes in soil available phosphorus in China (1990–2012). Field Crops Research, 192, 13–20.
Point 6: Bulk density is a critical factor, but how this was determined, either field or laboratory measurements is not stated. This needs to be addressed, the method justified, and any shortcoming discussed.
Response 6: Thank you for pointing out the missing details of bulk density. We have added more information about soil sample collection and laboratory measurements of bulk density in the Materials and Methods (Lines 141-143).
“Bulk density was determined from undisturbed soil cores collected in 100 cm3 cylinders after drying at 105°C for 24 h.”
Results
Point 7: The companion land management and nutrient input data has been sourced from the China Agriculture Yearbook (1992-2012). How this data matches the resolution of the actual field soils collected, could be discussed in more depth.
Response 7: Accepted and revised. We added more discussions in the new manuscript as follows (Lines 353-359):
“Statistical data could indicate the changes trend of nutrient input from chemical fertilizer, manure and straw return at the regional scale. Although the resolution of nutrient input data from statistical data did not perfectly match measurement data from actual field in our study, both datasets were collected based on farmers’ croplands. Thus, regional nutrient input data was a supplement of actual field measurements and could reflect the changes of famers’ field practices over time which provided an explanation for regional SOC changes.”
Point 8: Line 241. Grammar error. Correction needed.
Response 8: Revised as follows:
“Changes of SOCD under different soil types”
Point 9: Line 242. Here and throughout the text the way the two temporal categories are referred to can be improved for the purposes of readability. For example, “the 1991–2001 and the 2002–2012 varied”.
Response 9: We revised the sentence as follows (Lines 248-249):
“At the regional scale, the magnitude of the SOCD changes varied greatly across main soil types between the 1991–2001 and the 2002–2012.”
Point 10: Line 269. Rephrase. “had occurrences of 53–59% increase”.
Response 10: We revised the sentence as follows (Lines 300-303):
“Previous study reported that SOC in the topsoil of farmland in China (no data for Hainan, Tibet and Yunnan) increased in 53–59%, decreased in 30–31% and stabilized in 4–6% from the early 1980s to the late 1990s.”
Point 11: Define the DNDC model in full.
Response 11: Revised. We added the full name of DNDC (DeNitrification-DeComposition) model in Line 307.
Point 12: Line 286-289. The slight decrease mentioned doesn’t appear so insignificant when compared to the other regions where on average SOCD gains were made. The data for the NE should also placed in additional context taking into account the increase in SOCD for other regions.
Response 12: Revised. The reduction of SOC in NE region was analyzed in detail in the results section (Lines 226-228). The data in the NE region was added in discussion section in Line 346.
Point 13: Line 219-292. The positive correlation was found between the SOC stock and N fertilizer and crop residue C input [28]. Minor grammar inconsistencies. Sentence could be improved for purposes of better communication of the science.
Response 13: The sentence was revised as follows (Lines 276-278):
“The SOC stock was positively correlated with crop residue C, N fertilizer and organic fertilizer inputs.”
Point 14: Line 306. “that an increase of 472 Tg in C pool with the accumulative rate of 23.6 Tg yr-1”. Sentence could be improved for purposes of better communication of the science.
Response 14: We revised the sentence as follows (Lines 293-296):
“For example, Xie et al. [32] reported that the SOC stock in China’s croplands (156 Mha) increased by 472 Tg C, with an average rate of 23.6 Tg C yr-1 from the 1980s to the 2000s based on the Second National Soil Survey and published data.”
Point 15: Line 319-321. In fact, the more of the potential carbon storage budget that has been achieved already historically, the less of this budget that remains for future exploitation by improving management practices. This is an important point but could be articulated more clearly.
Response 15: Thank you for your very valuable comments. We have rewritten the sentence as follows (Lines 317-327):
“Previous reports indicated that there were great potentials for C sequestration under improved agricultural management strategies in China’s croplands (Qin et al., 2013; Tao et al., 2019). The SOC will increase asymptotically with additional C input and reach an equilibrium C level, especially for low SOC soil over a certain time, but the rate of increase of equilibrium SOC declines gradually without continuous C input indefinitely (Qin and Huang, 2010; Qin et al., 2013). The increasing C input no longer results in additional soil carbon when SOC at steady level reaches the ceiling (Qin et al., 2013). Thus, it is worth considering that the less of C budget would remain for future exploitation by improving management practices, especially for high SOC soils. These improved management practices efforts to increase SOC sequestration and to restore productivity should be further concentrated on the sites with lower SOC.”
References:
Qin, Z., Huang, Y. 2010. Quantification of soil organic carbon sequestration potential in cropland: A model approach, Sci. China Life Sciences, 53(7), 868–884.
Qin, Z., Huang, Y., Zhuang, Q., 2013. Soil organic carbon sequestration potential of cropland in China. Global Biogeochmical cycle. 27, 711–722.
Tao, F., Palosuo, T., Valkama, E., Mäkipää, R., 2019. Cropland soils in China have a large potential for carbon sequestration based on literature survey. Soil Tillage Research. 186, 70–78.
Point 16: Line 351. precise cultivation history led to the uncertainties in estimating regional changes of SOC. Minor grammar inconsistencies. Sentence could be improved for purposes of better communication of the science.
Response 16: The sentence has been revised as follows (415-419):
“Although we compared the SOC changes under different soil and crop types over two decades using large datasets, the uncertainties in estimating regional changes of SOC could be caused by the lack of more paired resampling data over time, detailed field management practices (e.g., fertilizer, crop residue, tillage, and irrigation) and precise cultivation history.”
Point 17: Line 355. laboratory analysis errors, but the huge soil samples were tested in different cooperating. Minor grammar inconsistencies. Sentence could be improved for purposes of better communication of the science.
Response 17: The sentence has been revised as follows (421-424):
“The same method/ procedure was used to reduce the deviation caused by the laboratory analysis errors, but huge soil samples were tested in different cooperating institutes under the IPNI program which could lead to uncertainties.”
Summary
Point 18: An important topic that would interest many, both within China and beyond. An extensive dataset with excellent spatial and temporal coverage. The findings are of high interest, although, they also largely support the trends in other works, the scale and scope of this study provide additional value.
Not all the relevant methods are described.
Response 18: Revised. We have provided all relevant methods in the new manuscript.
Point 19: There could be more discussion on how the reported trends relate to sink saturation, reversibility risk and leakage risk.
Response 19: We have added more discussion in the revised manuscript as follows (Lines 371-385):
“The increased SOC stock was observed in our study across most agricultural regions in China’s croplands, indicating that the existing SOC may still has a potential to increase mainly due to that the SOC stock has not reached the upper limit of the saturation in these regions (Qin et al., 2013). However, the decreased SOC stock in the NE region was detected, possibly because the new and lower equilibriums have not been reached yet due to a relative short cultivation history after conversion from forests and grasslands with high initial SOC (Yan et al., 2011; Zhou et al., 2019). It may reflect a great challenge of how to alleviate soil degradation in this region and achieve the C sequestration and emission reduction through effectively improved field management practices. Although CO2 leakage from carbon capture and storage was not analyzed in our study, recent study has shown that radial diffusion of CO2 leakage caused farmland degradation, degraded the morphological characteristics of spring wheat, and the impacts of CO2 leakage on horizontal topsoil CO2 concentrations was 2.67 times than that of the source depth (Ma et al., 2020). The SOC saturation, reversibility and leakage risk under different SOC levels at the regional scale should be considered in the future investigation.”
References:
Ma, X., Zhang, X., Tian, D., 2020. Farmland degradation caused by radial diffusion of CO2 leakage from carbon capture and storage. Journal of Cleaner Production, 255, 120059.
Qin, Z., Huang, Y., Zhuang, Q., 2013. Soil organic carbon sequestration potential of cropland in China. Global Biogeochmical cycle. 27, 711–722.
Yan, X.Y., Cai, Z.C., Wang, S.W., Smith, P., 2011. Direct measurement of soil organic carbon content change in the croplands of China. Glob Change Biology. 17, 1487–1496.
Zhou, Y., Hartemink, A.E., Zhou, S., Liang, Z., Lu, Y., 2019. Land use and climate change effects on soil organic carbon in North and Northeast China. Science of the Total Environment. 647, 1230-1238.
Point 20: One of the main conclusion messages was that more long-term field trials are needed. However, really how feasible is this? Do we have the luxury to wait for another 20 years for sufficient new data to be generated? What about advances in measurements that provide quicker indications of improved carbon capture? or better harnessing of existing field sites? This could be discussed. Also, the point about harnessing or building C in non-top soils could be explained in more depth. Does this refer to extending woodlands or hedgerows? Or trying to enhance C storage in deeper soil horizons.
Response 20: Agreed. We have revised these sentences to further point out the potential prospects for estimating the changes in SOC in cropland as follows (Lines 453-459):
“In order to better understand soil carbon cycling and to identify management practices which benefit C sequestration, it is important that combination of process-based models and long-term monitoring sites are more effective to estimate the response of SOC stock to different soil types and cropping systems under climate variability. Additionally, future research should focus on exploring the effects of various field management practices with potential to build C in deeper soil depths.”
Point 21: The manuscript still requires further proofreading and editing. My comments above are not exhaustive. Overall though I consider this work suitable for publication pending some further revision.
Response 21: Thank you for your very valuable comments. We have made a major revision for the manuscript by added more relevant methods and discussion in the main text.

Reviewer 2 Report
Authors of manuscript examined spatio- temporal variations in topsoil (0–0.20 m) SOC were analyzed using 43743 soil samples in China’s croplands. The results showed an increasing trend in SOC density (SOCD) for both grain and cash crops from 1991 to 2012.
However, the manuscript needs some major corrections
- The maps presented in the paper are land cover maps and not soil maps. Fig 1 is poorly described.
- If it is an international work, soil classification should be provided according to WRB (World reference for soil resources) or Soil taxonomy (USA).
e.g. pads soil - gleysols,
Meadow soil - what does it mean? There are no such soils in the WRB classification. In the meadow areas, there may be Black soil or another.
3.All figures and tables should be corrected in all manuscript according WRB or .Soil taxonomy
- 43,743 soil samples and not a word in the methodology about statistical analyzes. It is unacceptable for such a number of samples.
- Very contradictory discussion.
verse 186-188 The authors explain the increase in SOCD:
The overall trend of increased SOCD in China might be caused by the improved farm management practices. These included increases in chemical and organic fertilizer (human and live- stock waste) inputs.
The increase in chemical and organic fertilization will probably not contribute to reducing CO2 emissions. I completely disagree with it. Even the literature on the subject indicates that only balanced fertilization can produce such an effect. The results are discussed very briefly.
Author Response
Response to Reviewer 2 Comments
Authors of manuscript examined spatio- temporal variations in topsoil (0–0.20 m) SOC were analyzed using 43743 soil samples in China’s croplands. The results showed an increasing trend in SOC density (SOCD) for both grain and cash crops from 1991 to 2012.However, the manuscript needs some major corrections
Point 1: The maps presented in the paper are land cover maps and not soil maps. Fig 1 is poorly described. If it is an international work, soil classification should be provided according to WRB (World reference for soil resources) or Soil taxonomy (USA).
e.g. pads soil - gleysols,
Response 1: Revised. In this study, the Figure 1 was mapped based on the soil sample sites for both grain and cash crops. The main soil types have been showed in this figure. Additionally, we have revised the soil classification based on WRB (World reference base for soil resources), which was described in Materials and Methods (Lines 150-151).
Point 2: Meadow soil - what does it mean? There are no such soils in the WRB classification. In the meadow areas, there may be Black soil or another.
Response 2: Revised. We have revised to Meadow soil to Umbrisols based on the WRB classification.
Point 3: All figures and tables should be corrected in all manuscript according WRB or Soil taxonomy
Response 3: Accepted and revised. All figures and tables have been corrected according WRB in the new manuscript.
Point 4: 43,743 soil samples and not a word in the methodology about statistical analyzes. It is unacceptable for such a number of samples.
Response 4: Thank you for your very valuable comment. We have further explained the statistical analyzes in the revised manuscript. In this study, the minimum value, maximum value, standard deviation and coefficient variation were used to estimate the changes of SOCD (Lines 176-178 and Table 4). For significant statistical analysis, the resample sites from the 1991-2001 to the 2002-2012 in each region at same sites were used to analyze significant differences using the group T test at the 0.05 probability level (p < 0.05) in the SPSS 25.0 package (Table S1).
Point 5: Very contradictory discussion.
verse 186-188 The authors explain the increase in SOCD:
The overall trend of increased SOCD in China might be caused by the improved farm management practices. These included increases in chemical and organic fertilizer (human and live- stock waste) inputs.
The increase in chemical and organic fertilization will probably not contribute to reducing CO2 emissions. I completely disagree with it. Even the literature on the subject indicates that only balanced fertilization can produce such an effect. The results are discussed very briefly.
Response 5: We are sorry for the confusion. The sentence was revised, and more discussion was integrated in the revised manuscript as follows (Lines 270-273, 276-289):
“These included adoption of balanced fertilization, increased organic fertilizer (human and livestock waste) inputs and higher amounts of crop residues returned to soils for different geographical regions in China’s croplands between the 1991–2001 and the 2002–2012”
“Previous studies indicated that the SOC stock was positively correlated with crop residue C, N fertilizer, organic fertilizer inputs (Zhao et al., 2018; Berhane et al., 2020; Oladele and Adetunji, 2021). The SOC sequestration and CO2 emission reduction could be achieved by optimizing management practices (e.g., residue return combined with NPK fertilizer, organic manure application, and mixed application of organic and inorganic fertilizers) (Zhang et al., 2019; Liu et al., 2021). It has been reported that the SOC stock would increase by ≥ 25.0 Tg C yr-1 or 0.63% yr-1 through the implementation of the recommended management practices in China’s croplands, compensating for CO2 emissions by 1.0% (Tao et al., 2019). In addition, Zhao et al., (2018) further indicated that application of N fertilizer increased SOC at low SOC cropland, which was attributed to the increased crop dry matter production, resulting in more C inputs to soils. However, excessive chemical N fertilizer may constrain SOC sequestration through stimulating decomposition of cellulose-dominant crop residues, reduce C retention efficiency by favoring bacteria over fungi, and even limit root growth by stimulating soil acidification (Zhao et al., 2018).”
References:
Berhane, M., Xu, M., Liang, Z., Shi, J., Wei, G., Tian, X., 2020. Effects of long-term straw return on soil organic carbon storage and sequestration rate in North China upland crops: A meta-analysis. Global Change Biology. 26: 2686–2701.
Liu, B., Wang, X., Ma, L., Chadwick, D., Chen, X., 2021. Combined applications of organic and synthetic nitrogen fertilizers for improving crop yield and reducing reactive nitrogen losses from China’s vegetable systems: A meta-analysis. Environmental Pollution. 269, 116143.
Oladele, S.O., Adetunji, A.T., 2021. Agro-residue biochar and N fertilizer addition mitigates CO2-C emission and stabilized soil organic carbon pools in a rain-fed agricultural cropland. International Soil and Water Conservation Research. 9, 76-86.
Tao, F., Palosuo, T., Valkama, E., Mäkipää, R., 2019. Cropland soils in China have a large potential for carbon sequestration based on literature survey. Soil Till. Res., 186, 70–78.
Zhang, P., Xu, S., Zhang, G., Pu, X., Wang, J., Zhang, W., 2019. Carbon cycle in response to residue management and fertilizer application in a cotton field in arid Northwest China. Journal of Integrative Agriculture. 18(5), 1103–1119.
Zhao, Y., Wang, M., Hu, S., Zhang, X., Ouyang, Z., Zhang, G., Huang, B., Zhao, S., Wu, J., Xie, D., Zhu, B., Yu, D., Pan, X., Xu, S., Shi, X., 2018. Economics- and policy-driven organic carbon input enhancement dominates soil organic carbon accumulation in Chinese croplands. Proc Natl. Acad Sci., 115(16), 4045–4050.

Round 2
Reviewer 2 Report
The manuscript has been revised accordingly
Author Response
Response to reviewer 2
Point 1: The manuscript has been revised accordingly
Response 1: Thank you very much for your comments.